# Early Onset Colorectal Cancer: An Emerging Cancer Risk in Patients with Diamond Blackfan Anemia

**DOI:** 10.3390/genes13010056

**Published:** 2021-12-26

**Authors:** Jeffrey M. Lipton, Christine L. S. Molmenti, Pooja Desai, Alexander Lipton, Steven R. Ellis, Adrianna Vlachos

**Affiliations:** 1Division of Hematology/Oncology and Cellular Therapy, Cohen Children’s Medical Center, New Hyde Park, NY 11040, USA; pdesai4@northwell.edu (P.D.); avlachos@northwell.edu (A.V.); 2Feinstein Institutes for Medical Research, Manhasset, NY 11030, USA; cmolmenti@northwell.edu (C.L.S.M.); sashaliptoncool@gmail.com (A.L.); 3Donald and Barbara Zucker School of Medicine at Hofstra/Northwell, Hempstead, NY 11549, USA; 4Division of Epidemiology, Department of Occupational Medicine, Epidemiology and Prevention, Great Neck, NY 11021, USA; 5Department of Biochemistry and Molecular Biology, University of Louisville, Louisville, KY 40202, USA; steven.ellis@louisville.edu

**Keywords:** Diamond Blackfan Anemia, cancer predisposition, colorectal cancer, cancer screening and surveillance

## Abstract

Diamond Blackfan anemia (DBA) is a rare inherited bone marrow failure syndrome, the founding member of a class of disorders known as ribosomopathies. Most cases result from loss of function mutations or deletions in 1 of 23 genes encoding either a small or large subunit-associated ribosomal protein (RP), resulting in RP haploinsufficiency. DBA is characterized by red cell hypoplasia or aplasia, poor linear growth and congenital anomalies. Small case series and case reports demonstrate DBA to be a cancer predisposition syndrome. Recent analyses from the Diamond Blackfan Anemia Registry of North America (DBAR) have quantified the cancer risk in DBA. These studies reveal the most prevalent solid tumor, presenting in young adults and in children and adolescents, to be colorectal cancer (CRC) and osteogenic sarcoma, respectively. Of concern is that these cancers are typically detected at an advanced stage in patients who, because of their constitutional bone marrow failure, may not tolerate full-dose chemotherapy. Thus, the inability to provide optimal therapy contributes to poor outcomes. CRC screening in individuals over the age of 50 years, and now 45 years, has led to early detection and significant improvements in outcomes for non-DBA patients with CRC. These screening and surveillance strategies have been adapted to detect familial early onset CRC. With the recognition of DBA as a moderately penetrant cancer risk syndrome a rational screening and surveillance strategy will be implemented. The downstream molecular events, resulting from RP haploinsufficiency and leading to cancer, are the subject of significant scientific inquiry.

## 1. Introduction and Background

Diamond Blackfan Anemia (DBA) is a rare inherited bone marrow failure syndrome (IBMFS), the founding member of a class of disorders known as ribosomopathies. Most cases result from loss of function mutations or deletions in one of 23 genes encoding either a small or large subunit-associated ribosomal protein (RP) [1]. DBA is characterized by red cell hypoplasia or aplasia, poor linear growth and congenital anomalies of the skeletal, orofacial, genitourinary and other systems observed in 50% of patients with DBA [2] (Figure 1). While small cohorts and case reports documented the association of DBA with cancer [3,4,5,6,7,8,9,10,11], two recent analyses from the Diamond Blackfan Anemia Registry of North America (DBAR) have quantified DBA as a cancer predisposition syndrome of moderate cancer penetrance [12,13]. The most prevalent solid tumors presenting in young adults and in children and adolescents, are colorectal cancer (CRC; median age 41 years, range 20–51 years) and osteogenic sarcoma (OS; median age 11 years, range 4–34 years), respectively.

### 1.1. Proposed Carcinogenic Pathways in Diamond Blackfan Anemia

Several mechanisms have been proposed to explain red cell hypoplasia and aplasia and congenital anomalies resulting from selected RP haploinsufficiency. Ribosomal proteins, in particular RPL5 and RPL11, found in excess due to decreased ribosome biosynthesis, bind to HDM2 and block its activity as a ubiquitin E3 ligase, inhibiting the degradation of p53 [14]. The activation of p53 results in apoptotic death and cell cycle arrest in susceptible cells leading to cellular hypoproliferation [14]. Cell cycle perturbations due to RP haploinsufficiency may also be p53 independent [15]. Another proposed mechanism posits that the decrease in the number of ribosomes because of the loss of a critical RP ribosomal building block reduces translational capacity. This lack of sufficient translational capacity results in a global translational defect [1,16]. In a model of DBA in which zebrafish lacked Rps19 and exhibited failed erythropoiesis also resulted in significant developmental defects [2,17]. These lesions can be ameliorated by the activation of mRNA translation by the branched-chain amino acid L-leucine [18]. Furthermore, there is evidence that faulty ribosome biogenesis may also result in a selective decrease in translation of proteins critical for highly proliferative erythroid and other organ development. For example, the slow globin translation leading to the presence of non-globin bound heme creates a profound oxidative stress leading to erythroid cell death [19]. Furthermore, other proteins critical to erythroid and other tissue development such as GATA1 [20] and others [21], as well as non-erythroid tissue-specific factors [22] are subject to altered translation. Although it is not clear which mechanisms are dominant in a particular cell type there is ample evidence that RP haploinsufficiency results in decreased cellular proliferative capacity due to cell cycle perturbations, accelerated apoptosis or both.

De Keersmaecker and colleagues in several reviews [23,24,25] advance the concept that many human cancers can be considered acquired ribosomopathies. They point out that somatic heterozygous inactivating mutations or deletions observed in several genes encoding ribosomal proteins are associated with specific cancer types. These include relapsed and de novo T-cell acute lymphoblastic leukemia (T-ALL; *RPL5*, *RPL22*, and *RPL10*, respectively); gastric, endometrial and colorectal cancer (*RPL22*); multiple myeloma, melanoma, glioblastoma and breast cancer (*RPL5*) and others. Nucleolar stress is caused by the disruption of nucleolar structure and/or function due to a variety of factors often resulting in p53 activation [3,14]. Nucleolar stress in DBA is the result of abnormal ribosome biosynthesis and function due to haploinsufficiency of any one of several critical ribosomal proteins. As described, the unique role of p53 in the pathophysiology of DBA, in which the anti-proliferative effect of nucleolar stress from RP haploinsufficiency results in pro-apoptotic p53 activation leading to the DBA phenotype, may in turn lead to hyperproliferation. Although the science is not completely settled, it is hypothesized that this pro-apoptotic p53 activation drives selection for inactivating or compensatory somatic p53 mutations [26]. The concordance of RP haploinsufficiency and p53 loss of function (LOF) in cancer specimens and cell lines supports this premise. Further examination of tumor samples and animal models are underway to confirm this hypothesis. In addition, they and others show evidence that ribosomopathy-induced reactive oxygen species select for compensatory mutations that lead to hyper-proliferation by ameliorating anti-proliferative oxidative stress. Moreover, biosynthesis of altered ribosomes due to missing RP appears to create “onco-ribosomes” which favor the translation of growth-promoting oncogenes while disfavoring tumor-suppressors.

While the cause and effect of germline RP haploinsufficiency and somatic *Tp53* LOF mutations in DBA-associated cancer is yet to be determined, the concordance of cancer types in DBA (faulty ribosome biosynthesis) and the germline *Tp53* LOF-mutated Li-Fraumeni syndrome (LFS) as opposed to cancers observed in other IBMFS [3], Fanconi anemia (FA; faulty DNA repair) and dyskeratosis congenita (DC; faulty telomere maintenance), suggests that p53 LOF may contribute to cancer predisposition in DBA. However, the cancer penetrance in patients with LFS ages 16–50 years is much higher than for DBA; 51% for breast cancer, osteosarcoma, soft tissue sarcoma, leukemia, colorectal and lung cancer, as well as astrocytoma and glioblastoma [27], the latter two diagnoses not yet reported in DBA [12,13]. Clearly, the sporadic nature of Tp53 loss of function mutations in DBA as opposed to germline mutations in LFS could account for much of this difference. Nevertheless, we speculate that in addition to these mechanistic insights, these biologic and clinical similarities suggest lessons learned from strategies aimed at the mitigation of cancer risk in LFS may also be applicable to DBA.

Malkin and colleagues note “…that at least 10% of children with cancer harbor a pathogenic germline mutation in a known ‘cancer predisposition gene”, and have addressed the importance of screening and surveillance in those individuals at high risk in order to reduce morbidity and mortality through early detection [28]. Strategies are evolving. Effective cancer interception depends upon effective tools (molecular biomarkers, physical examination, imaging, etc.) that are highly sensitive while avoiding false positive tests leading to physical and psychological harm, as well as an accurate knowledge of the natural history of each disorder refined by sophisticated genotype–phenotype correlations. Armed with this information, chemoprevention, life-style modification and, when indicated, timely intervention once cancer is detected can mitigate the consequences associated with harboring a “cancer predisposition gene”. LFS, described in 1969 [29] and ultimately linked to germline mutations in *Tp53* [30], stands as a prototypic cancer predisposition syndrome in the evolution from discovery to effective intervention in a broad array of cancer types like those observed in DBA. As quantitative knowledge of the cancer susceptibility in DBA evolves so will screening and surveillance strategies using FFS as a guidepost. As a first step in a strategy to intercept cancer in DBA, we can take advantage of decades of work in the development of screening and surveillance in both sporadic and hereditary colorectal cancer to initiate a cancer screening and surveillance program in the context of the current state of knowledge regarding DBA as a cancer predisposition syndrome.

### 1.2. Cancer Incidence Data from the Diamond Blackfan Anemia Registry of North America 

The Diamond Blackfan Anemia Registry of North America (DBAR; www.clinicaltrials.gov (accessed on 23 December 2021), #NCT00106015) was established in 1991 and is the largest international DBA patient cohort with ongoing follow up. Three important studies from the DBAR defining the cancer risk and proposing a preliminary screening and surveillance strategy were published from 2012–2021 [12,13,31]. In 2018, Vlachos et al. [13] refined their 2012 study [12] using the DBAR cohort of 702 patients, median age 18 years, with 12,376 patient-years enrolled from 1991–2016. A competing risk approach was used to determine cause-specific hazard functions and cumulative incidence for the first adverse event (hematopoietic stem cell transplantation (HSCT), acute myeloid leukemia (AML), or solid tumor) as well as death from causes other than cancer (HSCT-related complications, iron overload, acute infection or sepsis, etc.) (Figure 1). Using the Surveillance Epidemiology and End Results Program (SEER 9; https://seer.cancer.gov/data (accessed on 23 December 2021)) data for an unselected population, the cancer risk in DBA was determined by comparing the observed cases to that expected for the population adjusted for age (observed to expected (O/E)). In this study, there were 39 cancers with 5 following HSCT. There were three cases of AML and 25 solid tumors in 26 patients not having undergone HSCT. Three squamous cell and three basal cell skin cancers were excluded from analysis as these malignancies are not included in the SEER data base. Although not a cancer, eight patients were diagnosed with myelodysplastic syndrome (MDS), a pre-leukemic disorder, at a median age of 26 years. Patients with DBA had a 4.8-fold higher relative risk (O/E), compared to the general population, of developing cancer with an overall cumulative incidence of 13.7% by age 45 years. An ominous harbinger is the steep hazard curve for solid tumors in DBA, rising from <1% to >5% per year between the ages of 35 and 60 years, with poor outcomes for those patients who develop cancer. The solid tumors with the greatest statistical risk, with more than a single case, were CRC and OS, with O/E of 45 and 42 respectively [13]. Five years later (DBAR median age 23.6 years for 813 patients), an update reported 51 patients with 60 cancers (10 post-HSCT and 7 basal or squamous cell skin cancers) with a median age of 33 years confirming and refining the earlier results (Table 1 and Table 2) [31]. The leading diagnoses remained the same, now with 11 predominantly early onset colorectal cancers (EOCRC; median age 41 years, range 20–51 years, 2 post-HSCT) and 7 cases of OS (median age 11 years, range 4–34 years, 3 post-HSCT). The crude incidence of CRC for non-post-HSCT patients was 2.5%, nine of 366 patients 20–51 years of age. Of the 298 patients in the DBAR with a known causative mutation the common DBA-associated genotypes were *RPS19* (41%), *RPL5* (14%), *RPS26* (11%), *RPL11* (8%), *RPL 35a* (8%), *RPS17* (6%), *RPS24* (4%), *RPS10* (2%) and other (6%). In 38 patients with cancer or MDS, for which there was a known DBA-associated *RP* mutation, those genotypes were represented at approximately the same percent as in the entire cohort, *RPS19* (39%), *RPL5* (16%), *RPL11* (18%), *RPL 35a* (16%), *RPS17* (8%), *RPS24*, *RPS10*, other (3%), except for *RPS26* which was absent in patients with cancer. While the numbers are too small to attach statistical significance to this finding, in the context of the near absence of *RPS26* haploinsufficiency in human cancers [23,32], the suggested absence of RPS26 haploinsufficiency in DBA-associated cancer is worthy of investigation. In addition, there are a considerable number of patients for whom a DBA-associated causative mutation is unknown (Table 1 and Table 2). This reflects the lack of complete genotyping in patients due, in large part, to diagnoses being made prior to the availability of complete diagnostic gene panels, the reluctance of insurance companies to cover genetic screening for germline mutations and the recognition of the relatively high frequency of gene deletions, missed by conventional sequencing approaches [33]. More data will emerge as future patients have access to expanded DBA-specific diagnostic gene sequencing panels.

In the setting of the near absence of RPS26 in human cancers, including those in the DBA patient cohort, it is worth noting some of the properties of RPS26 that set it apart from many of the other cancer-associated ribosomal proteins. There is increasing evidence on several fronts that ribosomes can be heterogeneous with respect to the complement of ribosomal proteins and that this structural heterogeneity leads to functionally distinct ribosome populations. One ribosomal protein linked to this heterogeneity is RPS26. RPS26 is known to arrive late to the assembly 40S subunit and plays a critical role in final steps involved in the maturation of 40S subunits [34]. Thus, heterogeneity for RPS26 presumably occurs after the initial maturation of 40S ribosomal subunits. In their studies on RPS26, Karbstein and her group have shown that ribosomes in yeast are heterogeneous with respect to the present or absence of RSP26. Intriguingly, ribosomes lacking RPS26 preferentially translate low abundance stress mRNAs more readily than ribosomes containing RPS26 and that difference reflects whether the mRNAs have optimal Kozak sequences (a particular nucleic acid motif in mRNA transcripts that promote the recognition of a start codon) or not [35]. This heterogeneity was proposed to be part of a physiological stress response with RPS26-deficient ribosomes produced by an unknown stress-induced release mechanism. Thus, the hypothesis that patients with *RPS26*-mutated DBA are less prone to develop cancer than patients with other RP mutations could be related to ribosome heterogeneity for RPS26. Like other ribosomal proteins involved in DBA, haploinsufficiency of RPS26 interferes with the maturation of 40S ribosomal subunits with the subsequent reduction of 40S subunits being responsible for the erythroid manifestations of DBA. If RPS26 were to function catalytically in 40S subunit maturation, it is possible that haploinsufficiency for RPS26 could result in an unusually high level of RPS26-deficient subunits in these patients. If these RPS26-deficient 40S subunits would then selectively translate subsets of mRNAs, it seems reasonable to hypothesize that some of these mRNAs may have a protective role in reducing cancer incidence in individuals with DBA haploinsufficient for RPS26. The confirmation of a less cancer prone genotype, RPS26-mutated, would influence any cancer screening or surveillance program for individuals with DBA. The observation must be confirmed, and the hypothesis should be tested.

### 1.3. Early Onset Colorectal Cancer and Diamond Blackfan Anemia

A recent review documents the increasing incidence of EOCRC, defined as colorectal cancer diagnosed prior to a patient’s 50th birthday [36]. Indeed, pediatric and medical oncologists working in the adolescent and young adult (AYA) space are being confronted with an increasing number of patients aged 20–39 years with colorectal cancer. In the United States, between 1974 and 2013 the incidence of colon and rectal cancer has increased 2.4% per year and 3.2% per year, for ages 20–29 years, and 1% per year and 3.2% since 1980 for those aged 30–39, respectively [37]. The cause is unclear but the majority of hypotheses center around diet, obesity, smoking, alcohol consumption, sedentary lifestyle and an altered microbiome. While germline genetic lesions predisposing to EOCRC are well described, with DBA being among the most recent, most cases of EOCRC appear to be sporadic in nature. Silla and colleagues suggest that EOCRC may be a different disease from late-onset colorectal cancer (LOCRC). EOCRC is more likely to be heritable. However, there is an overlap with sporadic disease in those less than 50 years of age. Three molecular pathways predominate in both sporadic and hereditary CRC: (a) chromosomal instability; (b) microsatellite instability; and (c) CpG island methylator phenotype [38]. The known associated hereditary syndromes account for approximately 2–5% of CRC and 16% of EOCRC [39]. The most common are familial adenomatous polyposis (FAP) first recognized as a distinct clinical entity in 1881 [40] and nonpolyposis Lynch syndrome (LS), first recognized in 1895 [41] representing chromosome instability and microsatellite instability, respectively [38,42]. FAP is an autosomal dominant inherited condition with about 30% new mutations in the gene encoding the tumor suppressor APC. It is characterized by multiple colorectal adenomas leading to adenocarcinoma, mostly in the distal colon. These patients are also at risk for upper GI cancer and other tumors. LS is an autosomal dominant condition resulting from mutations in genes encoding mismatch repair proteins (*MLH1*, *MSH2*, *MSH6*, *PMS2*) and is characterized by early onset, poor prognosis and predominantly right-sided colorectal cancers that are frequently synchronous or metachronous. These patients are also at risk for endometrial and ovarian cancer and other tumors. These and other syndromes have been extensively reviewed [42]. Additional germline mutations in cancer predisposition-associated genes not previously considered as CRC predisposition syndromes are being suggested as predisposing to CRC. The validity of these remains controversial, save LFS, for which early colonoscopy screening and surveillance is recommended. The suggestion of these additional predisposing germline mutations opens the possibility of unrecognized mutations [43] that may contribute to seemingly sporadic cases. Preceding the recognition of *RPS20* as a “DBA gene” [44], the presence of a multiplex family with CRC co-segregating with a mutation in the gene encoding RPS20, leading to haploinsufficiency, suggests the need to look for the presence of occult germline RP haploinsufficiency in the genesis of CRC.

As described, there is compelling evidence that EOCRC and LOCRC are distinct entities [38], varying regarding genetic predisposition, location, histopathology, molecular classification and the presence of more advanced disease and poorer outcome for EOCRC. As opposed to LOCRC, younger patients frequently present with disease in the distal colon and rectum with multiple lesions and unfavorable histology [45]. The poorer outcome in EOCRC is in part due to the absence, until recently, of a risk-based screening and surveillance strategy for those individuals younger than 50 years of age.

Comparing unselected non-genetic as well as genetic EOCRC to DBA-associated CRC will undoubtedly shed light on mechanisms of colorectal oncogenesis. Of the cases of CRC captured by the DBAR, the median age was 41 years, range 20–51 years. The crude incidence of EOCRC for non-post-HSCT patients was 2.5% (nine of 366 patients 20–51 years of age, eight less than 50 years of age). Two patients diagnosed post-HSCT, at ages 20 and 28 years and 2 and 15 years post-transplant, respectively, represented two of the three rectal cancers reported. Our data suggest that although the numbers are small, the cancer risk is higher in transplanted patients with DBA. In an NCI study, Alter and colleagues [3] report the risk of cancer is greater in transplanted vs. non-transplanted patients with O/E ratios of 81 vs. 2.5, 30 vs. 4.2 and 55 vs. 19, respectively for the IBMFS; DBA, FA and DC. Thus, the relatively high number of rectal cancers observed post-transplant is confounded by the possibility that they are related to HSCT [46] in patients already predisposed to CRC. The third patient with rectal cancer presented at the age of 49 years and had breast cancer 6 years earlier. Unfortunately, the DBAR was not designed to capture data regarding the location, histopathology and molecular classification of CRC. While lacking some important details, case reports demonstrate that more granular information needs to be gathered. Helpful examples include: a 33-year-old woman with genetically unspecified DBA reported with a tubular adenocarcinoma of the rectum with intra-and extramural spread and perineural invasion and no evidence of microsatellite instability [8]; a 16-year-old boy with genetically unspecified DBA with graft versus host disease diagnosed with a well-differentiated tubular adenocarcinoma of the rectum and multiple adenomas 12 years following a HSCT, with no evidence of microsatellite instability [9]; and a 41-year-old man with a germline *RPS19* mutation diagnosed with a well-differentiated adenocarcinoma of the sigmoid colon with widespread metastases, with no mutations detected in RAS/BRAF genes [10]. Granular data collection regarding demographics, lifestyle, family history, CRC location and histopathology, latency and other factors in patients with DBA and CRC are essential components of a screening and surveillance strategy. Emphasis must also be placed upon a detailed exploration of tumor genomics in order to describe the biology of tumorigenesis and progression in DBA-associated EOCRC.

### 1.4. A Colorectal Cancer Screening and Surveillance Strategy in Diamond Blackfan Anemia

Screening and surveillance for CRC in patients with DBA is modeled after the effectiveness of screening and surveillance colonoscopy in significantly reducing CRC-related morbidity and mortality in a population selected only by age as well as in those with CRC risk syndromes [47,48]. The approach is based upon principles established for non-hereditary and hereditary CRC, utilizing data specific to each. The DBA CRC screening and surveillance program is beginning implementation in the United States and Canada with the assistance and support from the Diamond Blackfan Anemia Foundation (https://dbafoundation.org (accessed on 23 December 2021)) and DBA Canada (https://www.dbacanada.com (accessed on 23 December 2021)).

In patients with DBA, CRC is generally well advanced with a poor prognosis when detected clinically. In addition, HSCT appears to increase the CRC risk. In common with other IBMFS with cancer predisposition, many patients do not tolerate full dose chemotherapy. Current data supports colonoscopy in adults beginning at the age when CRC risk is 0.6% or greater due to the improved outcomes with early detection. Thus, based upon the risk threshold of 0.6% or greater, guidelines for CRC screening and surveillance are evolving for LFS. Recent recommendations suggest that CRC screening in LFS should commence at 20 years of age, reduced from 25 years of age, based upon those recommendations [49]. Thus, we recommend implementation of guidelines for patients with DBA based upon current DBAR data. Nine de novo diagnoses of CRC ranging in age of diagnosis from 20 to 51 (median age 41 years, three diagnosed within the adolescent young adult age range (20–39 years)) with a crude incidence of EOCRC for non-post-HSCT patients of 2.5%.

The recommended follow up interval will be 5 years for those with normal colonoscopies. The gastrointestinal and oncologic society guidelines for surveillance colonoscopy for those with positive findings (colorectal neoplasia or cancer) [50,51,52] should then be followed. Two patients were diagnosed with CRC following hematopoietic stem cell transplantation (HSCT), 2 and 15 years post-transplant, ages 20 and 28, respectively. Therefore, in those patients transplanted prior to the age of 20 years, the first screening colonoscopy should be performed within one year of HSCT, if clinically feasible.

These recommendations will likely be modified as more patients are enrolled and the DBAR population ages, and as preliminary screening and surveillance approaches are evaluated. Once the latency from neoplasia to cancer is determined, modifications of age of first screening colonoscopy and the 5-year interval in surveillance may be required. Furthermore, possible genotype–phenotype correlations may alter the age for initial screening. It is of note that the DBAR has identified one patient who, at 8 years of age, had a colonoscopy due to unrelated abdominal pain and was diagnosed with a premalignant adenomatous polyp. Thus, quantitating the latency period from the earliest detectable premalignant lesions to CRC to guide future screening and surveillance will be a critical outcome from this preliminary approach. For example, in LS, recommendations were to start screening at 20–25 years of age with surveillance every 1–2 years until age 75 years. The screening interval of 1–2 years was based upon the short latency for the development of CRC with 20% of men and 7% of women developing CRC within 2 years of a previous colonoscopy [53]. However, genotype–phenotype correlations emerged allowing for modifications with exceptions for *MSH6* and *PMS2* gene carriers for whom colonoscopy now begins at age 35 years. Similarly, based upon clinical data for patients with FAP where screening starts in puberty (age 12–15 years), there is a follow up colonoscopy every 1–3 years depending upon the findings on each colonoscopy [54].

Based upon these data and contingent upon success of the CRC screening and surveillance strategy in DBA, it is likely that screening and surveillance will be extended to other DBA-associated cancers, next targeting OS, for which the benefits of such a program should outweigh the risks.

## 2. Summary and Conclusions

Although described in 1938 [55] and defined as an IBMFS in 1976 [56] Diamond Blackfan anemia is a relative newcomer as a genetic cancer risk syndrome. As a very rare disorder with only moderate cancer risk penetrance, it is only now that a cancer screening and surveillance strategy may be instituted. The increased risk of EOCRC in patients with DBA has been established and as the DBAR cohort ages more cases will be identified in all age groups. The screening plus surveillance approach in DBA is based upon the effectiveness of CRC screening and surveillance approaches in the general population as adapted to the highly penetrant cancer risk syndromes, such as Li Fraumeni syndrome, Lynch syndrome, and familial adenomatous polyposis.

Ongoing data capture through the DBAR will allow for the modification of the preliminary DBA CRC screening and surveillance strategy. The success in CRC will lead to expansion of the program for other DBA-related cancers as the DBAR matures providing the data necessary to refine and expand the approach. Plans include developing protocols approved to take advantage of an abundance of tumor and non-tumor colorectal tissue samples taken at the time of colonoscopy to facilitate important science, so that, in addition to extending and improving the lives of our patients, this rare ribosomopathy will provide an opportunity to further understand the biology of carcinogenesis.

## Figures and Tables

**Figure 1 genes-13-00056-f001:**
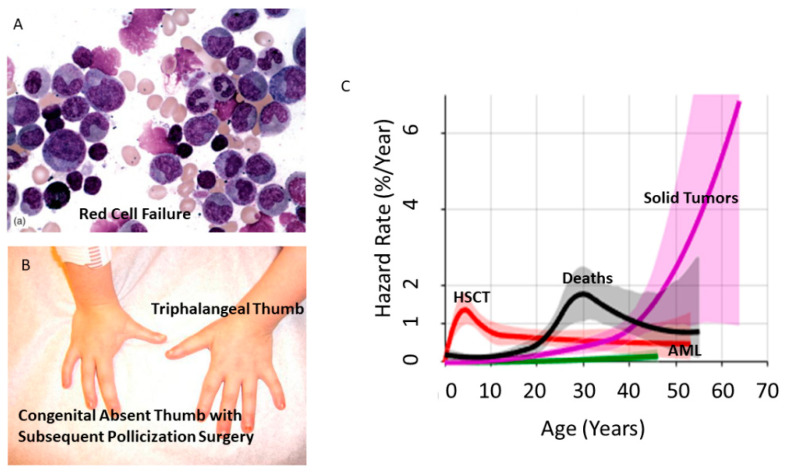
(**A**). Bone marrow aspirate demonstrating red cell aplasia, characterized by a selective absence of red cell precursors, in patients with DBA. (**B**) Typical radial ray anomalies: triphalangeal thumb and congenital absence of a thumb in a single patient. Image is post-pollicization surgery to construct thumb from the index digit. (**C**) Increased hazard rate for solid tumors beginning at age 20 years and increasing to 2% per year at age 45 years and continually increasing thereafter (purple). The hazard rate for acute myeloid leukemia (AML) starts to increase at age 45 years (green). The rate of deaths from other causes increased to over 1.5% per year at age 30 years (black) and the rate of hematopoietic stem cell transplant (HSCT) is highest under age 10 years with more patients undergoing transplantation at younger ages (red) [13].

**Table 1 genes-13-00056-t001:** Cancer, MDS or both in Patients with DBA (not having undergone HSCT).

Cancer Diagnosis	Number of Patients	Median Age at Cancer Diagnosis (Years; Range)	Genotype
**Gastrointestinal**	11		
Colon	9	41 (27–51)	*RPS19 (2); unknown (7)*
Gastroesophageal	1	28	*RPS17*
Esophageal	1	69	*RPL5*
**Genitourinary**	5		
Testicular	2	47 (32, 62)	*RPL35A; RPL5*
Uterine	1	64	*RPS19*
Cervical	1	27	*RPS19*
Squamous cell (vaginal)	1	45	*RPL11*
**Skin**	3		
Squamous cell (oral)	1	69	*RPL11*
Melanoma	2	38.5 (27, 50)	*RPL5; RPL11*
**Sarcoma**	5		
Osteosarcoma	4	17.5 (11–34)	*RPS19; unknown (3)*
Soft tissue	1	30	*Unknown*
**Hematological**	3		
Acute myeloid leukemia	3	44 (15–46)	*RPL35A (2); unknown (1)*
Non-Hodgkin lymphoma	1	41	*RPL5*
**Other**			
Breast	4	34 (22–43)	*RPS19 (2); RPS17 (1); unknown (1)*
Lung (small cell)	1	49	*RPS19*
Choroid meningioma of lung	1	21	*RPS19*
Wilms tumor	1	4	*RPL11*
**Myelodysplastic syndrome**	10	27 (1–61)	*RPL35A, RPS19, GATA1, unknown (7)*

**Table 2 genes-13-00056-t002:** Cancer post Hematopoietic Stem Cell Transplantation in Patients with DBA.

Cancer	Age at HSCT (Year)	Age at Cancer Dx (Year)	**Genotype**
Rectal carcinoma	13	29	*Unknown*
Rectal carcinoma	17	19	*Unknown*
Osteosarcoma	1	4	*Unknown*
Osteosarcoma	10	10 (4 months post-HSCT)	*Unknown*
Osteosarcoma	4	10	*Unknown*
Wilms tumor	6	8	*RPL35A*
Rhabdomyosarcoma	7	15	*RPS19*
Lung cancer	40 and 42	43	*RPL11*

## Data Availability

Not applicable.

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
