# Peer review of "Early Onset Colorectal Cancer: An Emerging Cancer Risk in Patients with Diamond Blackfan Anemia"

_genes, 2021, doi:10.3390/genes13010056_

Round 1

Reviewer 1 Report

This paper attempts to review broadly the pathophysiology of ribosomopathies and cancer risk, as well as cancer types associated with DBA, and ends with some general information for surveillance. It is not specific to CRC, though indicates this in the title. This is a very interesting question and area of focus, however the authors have struggled to write a cohesive and well-written article on this very broad topic. Some of the work referenced and reviewed in the paper already seems to have provided an excellent overview of many of the topics brought together here. I suggest narrowing down the focus to do a more indepth and novel analysis would provide the reader with more value.

Title: I would argue that DBA is a well- established cancer predisposition condition, increasing the risk of AML, MDS, and solid tumours (osteosarcoma, lung, and colon cancers). The ‘go-to’ info for geneticists is our Oxford textbook (2006) and Genereviews, both of which review the cancer risk with this condition. The risk of malignancy is between 2-5%, and cancer surveillance is part of the management of patients with DBA. I suggest re-wording the title and abstract to reflect that this is established, and not emerging information.

First paragraph: I suggest re-framing as cancer risk has long been known, and a more recent study confirms/further delineates the risks of cancer.

Line 40-41; for improved readability I suggest listing risks from youngest to oldest “…presenting in children is OS( median age, etc.) and in adults is CRC (median age) and include the actual risk of cancer here as well, the number of patients in the study etc. This will not be common knowledge to your readers, and will be important if the risk is higher or lower than historically believed to be, to add context to discussing surveillance later on.

Figure 1: Info below figures should related solely to figures, put discussion in the main body of the paper (ie. list of other congenital anomalies you have included in the description of the figure and surgical history of the patient-is the picture pre or post surgical correction?). Figure c: is this data from the previously mentioned study? The data is not from this paper, needs to be properly referenced at a minimum. Is this the rate of HSCT and deaths as stated, or the Harzard rate? And what is the difference?

Paragraph 1; I do not understand the proposed mechanism of action here, and suspect the authors may not either. Either remove most of it, or delve a bit deeper into the pathophysiology so that it is understandable to the reader, it appears that ‘key sentences’ have been taken from the reviews. A few detailed questions below for example

Line 62: how do erythroid failure and congenital anomlies result from cellular hypo-proliferation? This argument needs to be expanded to make sense to the reader. Or give other examples of congenital anomalies in other RPs to convince the reader.

Line 65: what is an interdicting mutation? And is this the normal process as opposed to when ribosomes have a mutation in them?

Line 67; this is not a complete sentence

Paragraph 2: I disagree that there is a striking concordance. LFS tumours are : brain, breast, sarcoma, and ACC. Leukemia is also seen, but it’s also the most common childhood cancer. CRC is more common in Lynch syndrome than LFS. And LFS is caused by a second hit to Tp53, and TP53 is very commonly hit in sporadic cancers. Your statement/premise would need evidence and numbers behind hit to support it.

Line 86’ is this a direct/personal quote? This reference is older, and not generally attributed to Dr. Malkin.

Line 90; what do you mean be preemption? Early detection? We are not pre-empting a diagnosis, we are catching it early with surveillance.

Paragraph 1.2; This first paragraph is a well-written comprehensive summary of previously published work from the registry.

Line 138: these are genes, not genotypes; genotype refers to the specific change in the gene.

144: how is this provocative? Provoking what? Complete your thought here.

Line 146: clarify that this is somatic cancers that have been sequenced that no RPSS26 mutations have been found Line 148-150-needs reference, what evidence?

Line 158: what is the relevance of Kozak sequences?

Line 161: resistant? Do they have less cancer than the general population, or do you mean do not have an increased cancer risk?

Paragraph 1.3 ; I think spate should be accompanied by a number or percentage increase, this is clearly not due to DBA, which does not have increasing incidence. You mention world wide, but then only give US numbers.

Line 224 how does HSCT increase the risk of CRC? Is this for all patients with HSCT or only those with DBA? Is the incidence of CRC higher in the DBA cohort than age matched population controls? This data would be appreciated to put the risk into context.

1.4; the lifetime risk of LFS for cancer is nearly 100% in women, using similar guidelines for a risk of <5% overall (or is it? I am not clear based on this paper), therefore the rationale for using this screening guideline is not clear to me, we do not do this for other cancer predisposition syndromes.

Line 255; ‘will begin’-do you mean in your clinic? If these are site-specific guidelines please clarify

Line 272: is this the first time Lynch syndrome is mentioned? If so, please spell it out, and probably explain what it is as well. Description of the lifetime risk of CRC in lynch compared to DBA would be relevant given you are outlining LS screening modifications based on gene involved.

Line 288; if a newcomer, please outline what year then this was accepted as a cancer predisposition condition, or when CRC was added.

Line 289-cancer surveillance has already been instituted, but not for CRC cancer. Pleae be specific about this.

Line 304- doesn’t taking an ‘abundance’ of tissue require REB’s, protocols, etc. It seems unusual this is added to the summary. What studies will be done on this tissue?

Author Response

Please see as attached file

Reviewer 2 Report

In this review, the authors focused on cancer incidence in DBA patients and recommended a screening and surveillance strategy for early onset colorectal cancer based on DBAR (DBA Registry of North America) data. The authors are experts on the clinical research of DBA and have accumulated the data from DBA patients for 30 years. This review certainly help people to gain updated cancer incidences in DBA and to understand importance of screening and surveillance strategies for colorectal cancer presenting in the adolescent and young adult DBA patients.

I was a bit surprised that there are many "Genotype unknown" cases for colon cancer (7 out of 9), osteosarcoma (3 out of 4) and MDS (7 out of 10) in Table 1. Any conceivable reasons for this? Is it possible that the mutated genes are not ribosomal protein genes? If so, what could be the relationship to ribosomopathies. I think it would be better to discuss this issue in the text.

Author Response

Please see as attached file.

Reviewer 3 Report

The manuscript is authored by experts in the field of Diamond-Blackfan anemia (DBA), who have contributed significantly to the study of this disorder. This review article provides a good source for clinicians and researchers, who interact with DBA at various levels. There are some observations on the manuscript:

  • This title of the manuscript is not clear whether the manuscript will provide a broad view of cancer predisposition in DBA or there is a focus on colorectal cancer (CRC). This is also reflected in the body of the manuscript and on Table 1, where there seems to be a discussion of all cancers in DBA with incidences and risk for their development.
  • The authors compare Li-Fraumeni syndrome (LF) to DBA and mention striking concordance of cancer types. Yet, there is no discussion about p53 mutation or protein overexpression in DBA. The statements are speculative without proving strong correlation regarding the p53 role in carcinogenesis in DBA.
  • The authors have not defined the age range for early vs. late onset CRC.

Minor issues:

  • Authors use the term “red cell failure” in the text and in Figure 1. The authors need to change it to either “red cell production failure” or “erythropoiesis failure”.
  • T-ALL is mentioned twice in the same sentence under 1.1. Proposed Carcinogenesis…
  • LOCRC is defined twice under 1.3. in the first paragraph and also in the second paragraph.
  • Under 1.2. Cancer incidence… The authors refer to Table 1 and Table 2, while there is only Table 1 provided.
  • For Figure 1, the hazard rate graph needs a reference if it is from a previous publication.

Author Response

Please see as attached file.
